# A single intra-articular dose of vitamin D analog calcipotriol alleviates synovitis without adverse effects in rats

Johanna A. Huhtakangas[1,2]*, Jere Huovinen[1], Sakari Laaksonen[3], Hanna-Marja Voipio[3], Olli Vuolteenaho[4], Mikko A. J. Finnilä[5], Jérôme Thevenot[5], Petri P. Lehenkari[1,6]

1 Cancer Research and Translational Medicine Research Unit, Medical Research Center Oulu, Oulu University Hospital and University of Oulu, Oulu, Finland, 2 Department of Medicine, Rheumatology Unit, Oulu University Hospital, Medical Research Center Oulu, Oulu, Finland, 3 Oulu Laboratory Animal Center, Department of Experimental Surgery, Oulu University Hospital and University of Oulu, Oulu, Finland, 4 Faculty of Biochemistry and Molecular Medicine, University of Oulu, Oulu, Finland, 5 Faculty of Medicine, Research Unit of Medical Imaging, Physics and Technology, University of Oulu, Oulu, Finland, 6 Division of Operative Care, Oulu University Hospital and University of Oulu, Oulu, Finland

* johanna.huhtakangas@kuh.fi

## Abstract

1,25-dihydroxyvitamin-$D_3$ and its derivatives have shown anti-arthritic and chondroprotective effects in experimental animal models with prophylactic dosing. The purpose of this preliminary study was to test the efficacy and safety of calcipotriol, vitamin D analog, as a treatment for a fully-developed knee arthritis in Zymosan-induced arthritis (ZIA) model. Forty 5-month-old male Sprague-Dawley rats were randomized into three arthritis groups and a non-arthritic control group with no injections (10 rats/group). A day after Zymosan (0.1 mg) had been administered into the right knee joints, the same knees were injected with calcipotriol (0.1 mg/kg), dexamethasone (0.1 mg/kg) or vehicle in a 100 µl volume. The left control knees were injected with saline (PBS) on two consecutive days. All injections, blood sampling and measurements were performed under general anesthesia on days 0, 1, 3 and 8. Internal organs and knees were harvested on day 8 and the histology of the whole knees was assessed blinded. Joints treated with calcipotriol showed a milder histological synovitis than those treated with vehicle (p = 0.041), but there was no statistically significant difference between the dexamethasone and vehicle groups. The clinical severity of arthritis did not differ between the arthritis groups measured by body temperature, swelling of the knee, thermal imaging, clinical scoring or cytokine levels on days 1, 3 and 8. Weight loss was bigger in rats treated with dexamethasone, propably due to loss of appetite, compared to other arthritis groups on days 2–3 (p<0.05). Study drugs did not influence serum calcium ion and glucose levels. Taken together, this preliminary study shows that a single intra-articular injection of calcipotriol reduces histological grade of synovitis a week after the local injection, but dexamethasone did not differ from the vehicle. Calcipotriol may have an early disease-modifying effect in the rat ZIA model without obvious side effects.

**Data Availability Statement:** All relevant data are within the manuscript and its Supporting Information files.

**Funding:** This work was supported by -the Finnish Medical Foundation https://laaketieteensaatio.fi/en/home/ (J.A.H.) -the Finnish Society for Rheumatology-https://www.reumatologinenyhdistys.fi/ (J.H.)

**Competing interests:** The authors have declared that no competing interests exist.

**Abbreviations:** GC, glucocorticoid; HE, hematoxylin and eosin; OA, osteoarthritis; ZIA, zymosan-induced arthritis.

# Introduction

Vitamin D and its natural metabolites have been shown to be major regulators of immunity and inflammation as well as bone and mineral metabolism [1,2]. Vitamin D signalling is crucial for maintaining joint health via the vitamin D receptor (VDR) which is highly expressed in all joint tissues such as synovium, cartilage, bone and muscle. Calcipotriol and its natural hormone 1,25-dihydroxyvitamin $D_3$ (calcitriol) have been shown to have similar anti-inflammatory and antiproliferative effects *in vitro* [3,4], but calcitriol has 100–200 times higher potency on calcium metabolism [5].

1,25-dihydroxyvitamin-$D_3$ and its derivatives have shown anti-arthritic and chondroprotective effects in experimental animal models. Preventive dosing of 1,25-dihydroxyvitamin-$D_3$ and its derivatives suppressed collagen-induced arthritis or restored joint function in rats in all [6–8] but one study [9]. In accordance with this, prophylactic treatment with vitamin D molecules have also shown disease-modifying effects in some surgical models of osteoarthritis (OA) in rodents in the early stages of OA [10–12] but this initial beneficial effect was lost in the advanced stages of OA [10–11].

Zymosan, a polysaccharide derived from the cell wall of yeast, is a natural ligand of toll-like receptor 2 (TLR2) [13] and Dectin-1 [14] that strongly activates the components of innate immunity including complements and mono- and polymorphonuclear cells [15]. The acute clinical arthritis with an increased blood flow and leucocyte infiltration is transient lasting approximately 1–2 weeks. The acute phase is followed by a chronic synovitis leading to gradual structural damage of cartilage and bone starting after a week [16–18].

Glucocorticoids (GC) have been the golden standard for the local treatment of arthritis due to their efficacy on inflammatory symptoms (pain and swelling), but they have well-known metabolic adverse effects. Regular use of local triamcinolone every three months for two years led to greater cartilage volume loss and no beneficial effect on knee pain compared to saline injections [19]. In a recent cohort study, there was a possible association between using GCs in knee OA and higher risk of disease progression [20].

The aim of this study was to test hypothesis that calcipotriol that is shown to have anti-inflammatory effects on synoviocytes *in vitro* [4] would also exert antiarthritic influence *in vivo* in rat Zymosan-induced arthritis (ZIA) model. We pursued the treatment response of fully-developed arthritis with a single intra-articular injection of calcipotriol, mimicking a clinical situation, instead of administering the compound in a prophylactic manner as in earlier studies. The second aim was to investigate the safety of a local calcipotriol injection during acute arthritis when increased systemic absorption of the compound is likely to take place due to vasodilatation.

# Materials and methods

## Animals

The study protocol was reviewed and approved by the National Animal Experiment Board of Finland (licence numbers ESAVI/5798/04.10.07/2017 and ESAVI/1878/2018/2/2018). The animal care and experimental procedures were in line with the Finnish legislation and EU Directive 2010/63/EU. The reporting of animal experiments in this study is in compliance with the ARRIVE guidelines [21].

Altogether 54 male outbred Sprague-Dawley rats of this study were housed at the Oulu Laboratory Animal Center of Oulu University at standard conditions (room temperature of 21 ±1˚C, relative humidity 40–60%, ventilation rate 15 ACH, illuminance 350 lx at 1 m height

and a 12/12 hour light-dark cycle with a gradual one-hour brightening and dimming). After weaning, two individual rats were housed together.

Their diet was standard chow (Teklad Global 18% protein rodent diet, Envigo) that contained 1.5 IU/g of vitamin $D_3$, 1% calcium and 0.7% phosphorus. Untreated municipal tap water and food were available *ad libitum*. Fresh slices of cucumber were used as an enrichment food.

## Study compounds

Zymosan (Cayman chemical, Ann Arbor, MI, USA), Dexamethasone with ≥99% purity (Merck, Kenilworth, NJ, USA) and calcipotriol with >98% purity (Santa Cruz Biotechnology, Dallas, Texas, USA) were stored at -20˚C until use. Dexamethasone and calcipotriol were prepared by mixing with 10 mg/ml lidocaine, (Orion, Espoo, Finland) and 99% ethanol containing the predetermined amount of the steroid hormone in a 96/4 volume ratio (lidocaine/ethanol) just before injections. The vehicle contained only lidocaine and ethanol in a 96/4 volume ratio without steroid hormones.

## Determination of Zymosan dose and study design based on a pilot study

A pilot study was carried out with fourteen eight-week-old male Sprague-Dawley rats (weight range 389–477 g) to evaluate the suitability of this Sprague-Dawley stock for ZIA (no previous scientific reports were available). A Zymosan dose of 0.1 mg was high enough to trigger an acute clinical arthritis in Sprague-Dawley rats based on a clinical and histological examination of the synovial samples from days 1, 3, 6 and 8. Initially, a dose of 0.5mg was administered in the pilot study to two rats, but the arthritis was considered clinically too severe for the purposes of the experiment and considering the 3Rs principle.

Based on these findings the main endpoint in the main study was defined to be a synovitis score of the femoral and tibial component separately and together on day 8 when synovial proliferation had been developed. The secondary endpoints were clinical treatment response on day 3 measured as a relative decline in fever, surface temperatures of the knees, swelling of the joint, cytokine levels and disability (clinical scoring).

## The study design and protocol of the main study

Forty 5.5-month-old male Sprague-Dawley rats (weight range 501–727 g) were randomized into three arthritis groups and a non-arthritic control group. On day 0, Zymosan (0.1 mg) was injected into the right knee joint. On day 1, the rats in the arthritis groups were injected with calcipotriol (0.1 mg/kg), dexamethasone (0.1 mg/kg) or vehicle in a 100 µl volume into the right knee joint in a random order and blinded by two persons giving injections. The left control knee was injected with saline (PBS) on two consecutive days (days 0 and 1). On day 8, the rats were euthanized and whole knees and internal organs were harvested.

All procedures were done in the same order under isoflurane anesthesia (VetTech Solutions Ltd, Congleton, United Kingdom UK) on days 0, 1, 3 and 8 for the three arthritis groups and under terminal anesthesia for non-arthritic controls (only on one day).The study protocol included shaving of the skin areas on both knees and abdomen (day 0 only), moisturizing eye drops, measuring of the rectal temperature with a thermometer (Thermalert TH-5, Physitemp, Clifton, NJ, USA), a standardized set of thermal images (4 different positions), whole body photographs, applying a thermal pad to prevent a heat loss, prewarming the tail by dipping it into 40˚C water for one minute, blood sampling from the tail vein (days 1, 3 and 8) with a 20G x 1½ vacuum needle (Terumo Venoject, Leuven Belgium), measuring the medio-lateral thickness of both knees at the level of the joint gap with a digital caliper (Mitutoyo, Sakado, Japan),

medial knee injections with 25 G needles, dosing of systemic analgesic (Buprenorphine, Vetergesic Ceva Santé Animale, Libourne, France) with a recommended dose of 0.05 mg/kg twice a day regularly until the evening of the second postoperative day, and euthanasia by decapitation under deep anesthesia (3–5 min of 4% isoflurane) on day 8.

Calcium ion, electrolytes, blood gases, glucose, hemoglobin levels were immediately measured with an I-stat handheld blood analyser (Abbott, Abbott Park, Illinois, USA) from whole blood and serum samples for cytokines and 25-hydroxyvitamin-$D_3$ were stored at -20˚C before analysis with a chemiluminescence based method.

## Daily clinical evaluation of rats

Clinical scoring and weighing were done every morning. We used a composite score including use of legs, pain reaction when compressing the arthritis knee lightly with fingers, activity in cage, fur appearance, facial expressions and vocalizations, which were evaluated by an animal caretaker every morning throughout the experiment (Supporting Information S1). Animal welfare was checked several times per day and systemic analgesic was dosed by animal caretakers. The criteria for euthanasia (humane end points) were determined beforehand. Criteria for euthanasia: The rat does not move or eat for 24 hours despite the analgesic treatment, the rat cannot carry food in its front paws or has to lean on its tail for support despite the analgesic treatment, the rat vocalizes abnormally or expresses other pain-related behaviour that does not respond to analgesics within 24 hours, the rat experiences maximal swelling of the joint (+50% change in perimeter) with pain despite the analgesic treatment, and the situation remains unchanged for 24 hours, the rat loses its weight more than 20% of the initial weight.

## Thermal imaging

All procedures were conducted in a windowless, temperature-controlled room. The thermal camera (Flir T420 thermal camera, FLIR Systems, Oregon, USA) and the rats were always positioned with the same distance between them by using markings on the table and floor. The variation in possible heat loss due to anesthesia was attempted to be minimized by measuring the heat-related variables as quickly as possible.

A custom-made algorithm was developed in MATLAB (v. 2017b, Mathworks, Natick, MA, USA) to perform manual segmentation of the regions of interest (ROIs) and analyse the thermal information within them. The images were coded randomly to ensure the blinding of the person drawing the ROIs.

The average, maximum and minimum temperature values were then calculated within each ROI. Furthermore, the homogeneity of thermal values ($H_{temp}$) was extracted to describe the spatial distribution of the temperature values and quantify the overall "smoothness" of the thermal distribution within each ROI. A thermal-level co-occurrence matrix (TLCM) was first calculated with rotational invariance to assess the spatial distribution of the thermal values. The homogeneity of the temperatures was then derived from the TLCM as follows:

$$H_{temp} = \sum_{i,j} \frac{TLCM(i,j)}{1 + |i - j|} \tag{1}$$

where each element (i,j) in the TLCM is the sum of the occurrences that a pixel with thermal value i has a pixel with thermal value j within a one-pixel radius. The pixel levels within the TLCM were grouped every 0.2˚C, based on the accuracy of the thermal camera. Eventually, a high value of $H_{temp}$ corresponds to ROI with a very "smooth" distribution of temperatures, meaning that pixels within the ROI have similar temperature values to their surroundings. In the chosen position, a ROI was drawn by a round-shaped tool on both knees and abdomen

(reference area). In the knee area, a ROI was selected covering the relevant joint structures as consistently as possible. Clearly colder areas due to leftover fur as well as pixels at the border of the image were avoided due to thermal diffraction. The average values were used in the statistical analysis.

### Preparation and analysis of the histological specimen

After euthanasia the heart, lungs, kidneys, spleen and both knees were collected and immediately placed in 10% neutral buffered formalin. The whole knees were demineralized in neutral buffered 10% EDTA and 4% formaldehyde for 8 weeks with buffer changes twice a week. Decalcified whole knees were embedded overnight with paraffin (Tissue Tek, SAKURA, Torrance, CA, USA), were cut into 5μm sagittal slices with a microtome approaching from the medial side to midline (Micron AM 355S, Thermo Scientific, Walldorf, Germany), stained with hematoxylin and eosin (HE) and toluidine blue, scanned into a digital format with 1:20 enlargement and evaluated with QuPath (version 0.1.2) [22]. The synovial grading was done in a blinded fashion from 0 to 4 as previously described [23].

The analysis method for synovia was validated with another reader who scored ten randomly picked samples independently.

### Statistics

Power analysis was conducted to estimate the minimum number of animals that would enable reaching statistical significance for the difference between the arthritis groups. Assuming 30% reduction in the primary end point between therapeutic and vehicle groups at the significance level of 0.05 and desired power of 0.80, we ended up with the minimum of nine animals per group using SPSS (version 26, IBM, Armonk, NJ, USA).

The primary endpoint (synovial scoring) and secondary end points (fever, local temperature of the knees, knee swelling) were studied by using Mann-Whitney U-test due to the small number of subjects (n = 8–10/group after exclusions) and ordinal scale of the data in synovitis score. The efficacy of calcipotriol on the synovitis score was compared directly against vehicle for the femoral and tibial components together and separately, as it was hypothesized that calcipotriol might reduce these parameters compared to vehicle.

In safety-related measurements (blood glucose, calcium-ion) or the measurements with no definite hypothesis (such as cytokines), the differences between the groups were analysed with either one-way ANOVA or Kruskal-Wallis test, depending on the distribution of the data. The increase in surface temperature and knee calibers between days 0 and 1 were studied with paired-samples t-test as all arthritis groups had similar treatment until day 1. The difference of average surface temperatures and knee calibers between right and left limbs for each arthritic rat was calculated on day 1, and these values were tested against zero in one-sample Wilcoxon signed rank test (nonparametric data).

## Results

All but one of the forty rats elicited clinical signs of arthritis by the following morning (within 20 hours) after an intra-articular Zymosan injection. One rat developed a large injection site hematoma and progressive anaemia (Hb 68) after Zymosan administration. Two other rats developed bowel obstruction and aspiration (lung tissue was filled with granulocytes). Both rats elicited monoarthritis of the right knee, which makes it unlikely that the injections had gone directly to circulation, ending up in lung tissue. Eating bedding material and an acute ileus are well-known side effects of opiates. After exclusions of these four rats before treatments, the final number of rats was reduced to thirty-six; ten in control group, eight in

A

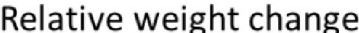
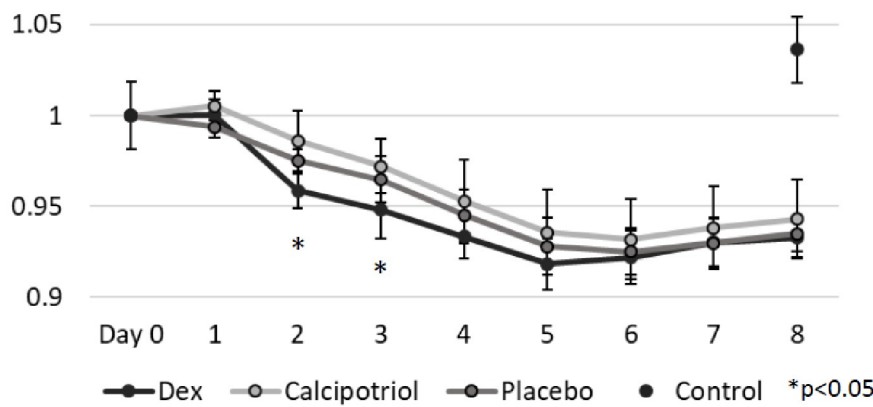

B

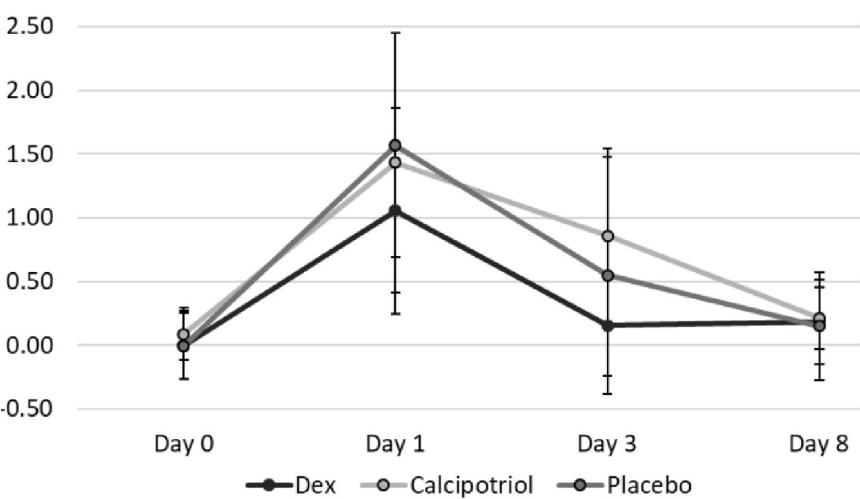

**Fig 1.** The relative average body weights (A) and knee joint swelling (right-left knees) of the arthritic rats during the experiment. Zymosan was injected on day 0 and the study compounds on day 1. (A) The relative weight reduction ((day X-day 0)/100) was significantly greater in the dexamethasone group than in the vehicle group on days 2 and 3, (p = 0.003 and p = 0.02 respectively). (B) The effect of the study compound on knee joint swelling was measured as a difference in the medio-lateral diameter at the joint gap level between the right arthritic and the left control knee. The severity of the clinical arthritis measured by swelling did not differ between calcipotriol and vehicle groups or dexamethasone or vehicle groups on day 3 or 8 (p = ns). The non-significant difference on day 1 is likely due to chance as study compounds were administered after the measurement. The number of rats was eight in calcipotriol and vehicle groups, nine in dexamethasone group, and ten in the control group. The data are expressed as mean ± SD.

**Table 1. Clinical and laboratory values of rats in different arthritis groups at baseline (day 1) and at termination.**

|  | vehicle n = 8 | dexamethasone n = 9 | calcipotriol n = 8 | control n = 10 |
|---|---|---|---|---|
| timing | day 1 | day 1 | day 1 | at termination |
| body temperature (˚C) | 37.7±0.6 | 37.6±0.4 | 37.8±0.6 | 37.2±0.5 |
| right knee diameter (mm) | 12.4±0.7 | 11.9±0.8 | 12.1±1.0 | 10.0±0.3 |
| weight (g) | 636±62 | 625± 54 | 613±72 | 615±57 |
| blood glucose (mmol/l) | 10.6±0.6 | 10.5±0.6 | 10.6±1.7 | 11.0±1.8 |
| Ca-ion (mmol/l) | 1.14±0.06 | 1.18±0.08 | 1.20±0.10 | 1.18±0.09 |

An intra-articular Zymosan injection was given on day 0. The corresponding data of non-arthritic control rats were taken at termination. The body temperatures (rectal), right knee diameters, weights as well as glucose and calcium ion levels were comparable between all arthritis groups (p = ns) before injection of study compounds (day 1). The data are expressed as mean ± SD.

calcipotriol group, and nine in dexamethasone and vehicle groups. No rats had to be euthanized due to the humane end points (Supporting Information S1).

## Clinical characteristics

The weight of the rats at the beginning of the experiment ranged from 501 g to 727 g, with no significant differences between the groups (Fig 1A, Table 1). All arthritic rats lost weight during days 1–6, and the body weight was at its lowest on days 5–6 (maximal relative weight loss -7-8% from beginning in all arthritic groups) while non-arthritic rats group gained +7.2 g/ 1.2% weight in average during the experiment. The relative weight loss was greater in the dexamethasone group than in the other arthritis groups on days 2–3 (6% vs. 3–4%, respectively, p<0.05) (Fig 1A).

Knee joint swelling was expressed as a difference in medio-lateral diameter between the right arthritic and left control knees (Fig 1B). The width was calculated as the average of three consecutive measurements at joint gap level. The average difference between the right and left knees on day 1 was 1.3 mm ± 0.89mm (p<0.0005). The subtraction diameters of the transverse measurement (right-left knee) did not differ on days 0, 1, 3 and 8 across the arthritis groups (Fig 1B).

There were no statistically significant differences in blood Ca-ion nor glucose levels at baseline (day 1) or on day 3 or 8 between the groups (Table 1, Fig 2A and 2B). Serum 25-$(OH)_2D_3$ levels, reflecting the nutritional status of rats, did not differ between the study groups measured at the end of the study (45±10.0, 49±8.6, 44±6.4, 48±6.0 nmol/l in vehicle, dexamethasone, calcipotriol and control groups, respectively) (p = ns). 1,25-dihydroxyvitamin-$D_3$ levels were not measured since the chemiluminescence-based method interferes with vitamin D analog calcipotriol (hydroxyl group positions 1 are the same).

The clinical scoring of the arthritic rats showed no significant differences across groups throughout the study (days 0–8) (Supporting Information S2). The rats were able to move and eat normally and no pain behaviour was observed during the experiment.

## Thermal imaging

The position where the hind legs were pulled into alignment with the body with soles touching showed the best homogeneity values (the best feasibility) in thermal imaging and was selected for analysis (Fig 3A). The average surface temperatures of either right or left knees did not differ across the arthritis groups at any time points during the study (Fig 3B). The increase in average temperature of the right knee from day 0 to day 1 was 0.74˚C (from 35.02 to 35.76˚C, p<0.0005) that reflects vasodilatation caused by inflammation. A smaller elevation of the

A

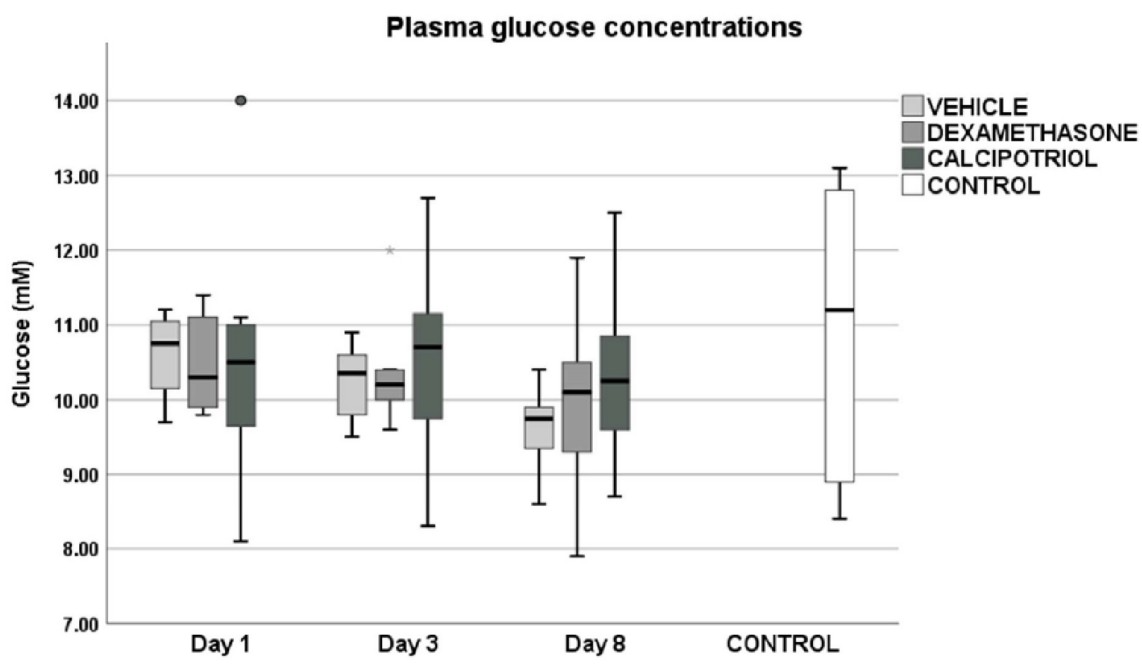

B

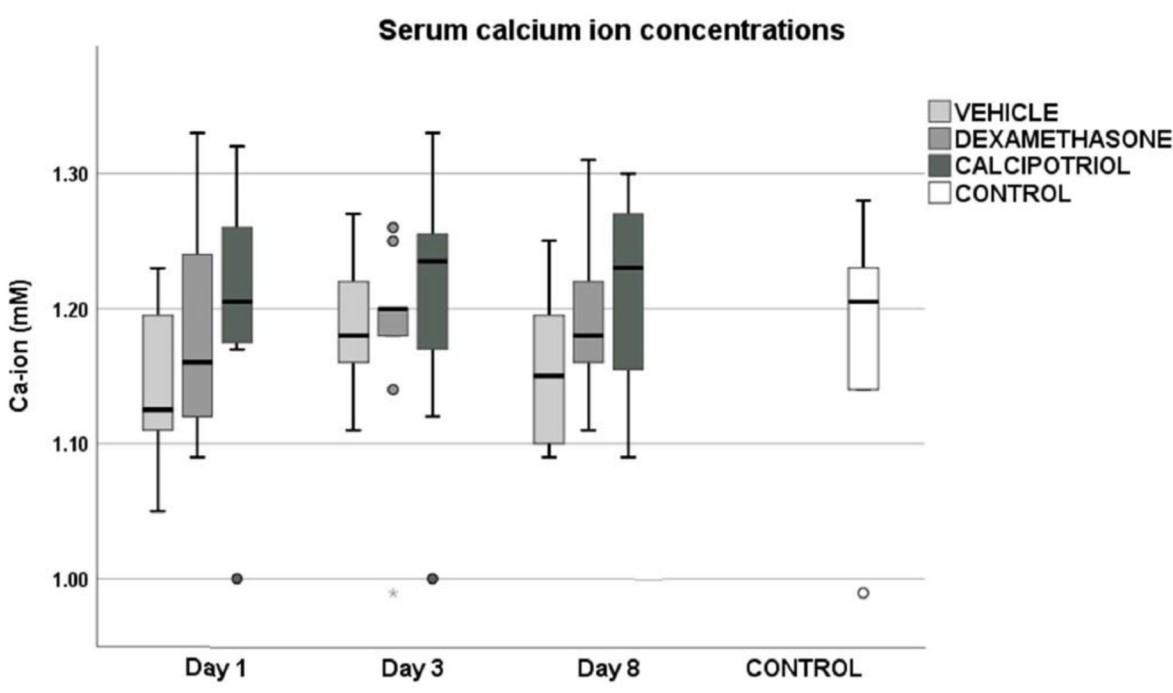

**Fig 2.** The plasma glucose (A) and serum calcium ion concentrations (B) of the rats as boxplots. Dexamethasone did not raise glucose levels and calcipotriol did not affect calcium levels significantly between days 1 and 3 or days 1 and 8. No hypercalcemia was observed in any rat. Note that the higher median concentration of calcium ion in the calcipotriol group on day 1 is likely due to chance as the samples were taken prior to dosing of the study compounds. Outliers that are over 1.5 times the height of the box are marked as grey dots in boxplots. The number of rats was eight in calcipotriol and vehicle groups, nine in dexamethasone group, and ten in the control group.

average temperature, 0.59˚C, was seen in the saline injected left knees (from 35.04 to 35.63˚C, $p < 0.0005$) that is caused by the injection itself (on day 1 right vs. left knee difference $p = 0.017$ in one-sample Wilcoxon signed-rank test compared against zero). No significant increase in the surface temperature of the abdomen (from 34.97 to 35.09˚C, $p = 0.356$) was detected. The elevated surface temperatures were maintained on day 3, but were reduced to baseline by day 8. (Fig 3B).

## Systemic inflammation

Rectal temperature was used as a measure of systemic inflammation. Baseline values were approximately the same, 37.1±0.5˚C on average in all arthritic rats pooled (Table 1, Fig 3C), and the temperatures raised by 0.6˚C on average (37.7±0.5˚C in arthritic rats) and reached their highest levels on day 3 (37.9±0.6˚C in arthritic rats) (Fig 3C). Temperature changes between days 1–3 and 1–8 were similar across the arthritis groups ($p = ns$). The body temperatures returned to baseline on day 8 and were comparable with controls (37.1±0.4 in the arthritic vs. 37.2 ±0.5˚C control group).

## Scoring of the histology of synovia from the whole knee specimen

Synovitis scoring of right tibial and femoral components together (total knee), as well as tibial site alone showed lower average scores of synovitis in the calcipotriol group compared to the vehicle group ($p = 0.041$ for total knee and $p = 0.033$ for tibial part), but there was no statistical significance in the average scores at the femoral site alone ($p = 0.091$) (Fig 4A). In contrast, the average scores in the dexamethasone group did not differ significantly from the vehicle ($p = ns$). Histological grades of the left knees of arthritic rats did not differ from that of controls (Fig 4B). Histology of the whole knees showed a gradual increase in proliferation of the synovial membrane according to the arthritis group (calcipotriol < dexamethasone < vehicle) while the surface layer of synovium in controls was 1–2 cells thick (Fig 5A–5H).

Cartilage erosions were mostly absent in the samples and extensive cartilage damage was not observed.

## Cytokine data

Zymosan caused systemic inflammation with a marked elevation in multiple pro- and anti-inflammatory cytokines TNF-α, IFN-γ, IL-13, G-CSF, M-CSF, IL-6, IL-18, IL-1α and IL-4 in arthritic rats (all arthritis groups pooled) versus the non-arthritic controls ($0.001 < p < 0.05$) (Fig 6). The level of RANTES (CCL5) was reduced on day 1 in arthritic rats versus controls ($p = 0.002$), which may indicate increased consumption. The levels of inflammatory cytokines were highest on day 1 after which they spontaneously declined in all groups irrespective of treatment on days 3 and 8. There were no statistically significant differences in the changes in cytokine levels across the arthritis groups, except for VEGF level which declined the most in the calcipotriol group during days 1–8 ($p = 0.046$), but this difference was not significant in pairwise comparisons.

A

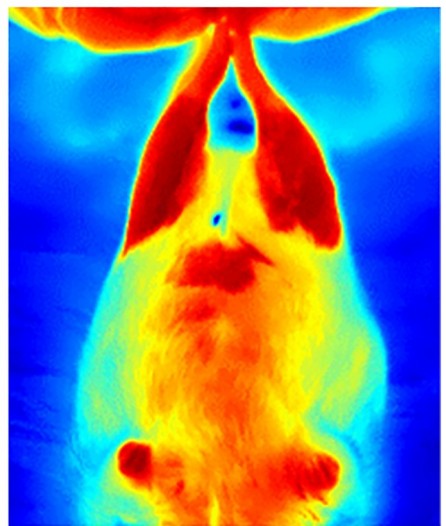

B

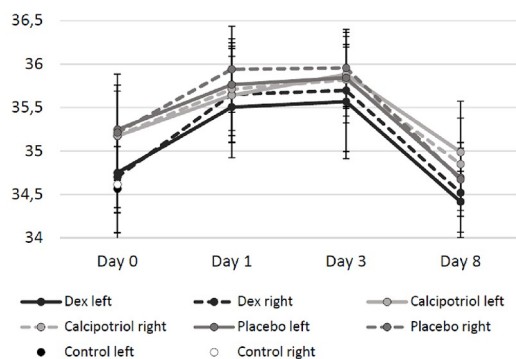

C

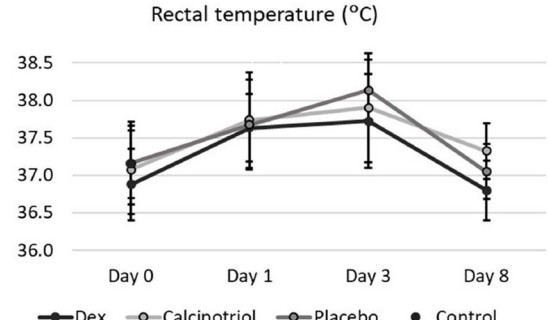

**Fig 3.** Position of the rats during thermal imaging (A), the average surface (B) and body temperatures (C) of the rats during the experiment. For thermal imaging (days 0, 1, 3 and 8) the optimal position was selected based on the highest homogeneity values. (A) The skin areas on abdomen (reference) and legs were shaved on day 0. The deeper is the red, the higher is the surface temperature. (B) Zymosan induced arthritis clearly elevated the surface temperatures of the knees, with a significant temperature difference between right and left knee on day 1 in arthritic rats (p = 0.017), but were no significant differences in the surface temperatures across the three arthritic groups after treatments on days 3 and 8. The surface temperatures of the knees that were measured by thermal imaging followed the curves of the body temperatures (rectal) (C) Also, the changes in body temperature or the surface temperatures from day 1 to 3 or from day 1 to 8 were comparable in calcipotriol vs vehicle and dexamethasone vs vehicle groups (p = ns). The body and surface temperatures returned to baseline on day 8. The data are shown as mean ± SD. The number of rats was eight in calcipotriol and vehicle groups, nine in dexamethasone group, and ten in the control group.

## Discussion

This preliminary study suggests that a single intra-articular calcipotriol injection 0.1 mg/kg alleviates synovitis in the ZIA rat. Calcipotriol prevented thickening of the synovial membrane by inhibiting cell proliferation which is reflected as lower synovitis scores in calcipotriol versus control group on day 8 (p = 0.041). In contrast, dexamethasone with a regular used dose of 0.1 mg/kg did not show a significant beneficial effect on the synovitis scores in this study. This is supported by our previous *in vitro* studies showing that calcipotriol has a stronger antiproliferative effect than dexamethasone on synoviocytes at equal doses [4]. The weight loss in general in all arthritis groups, is most probably due to loss of appetite caused by inflammation, stress and pain, and further enhanced by dexamethasone itself. Dexamethasone caused significantly more weight reduction in two days after administering the drug compared with vehicle and calcipotriol, probably due to decreased appetite, as the weight loss was rapidly recovered after two following days. Previously intraperitoneal dosing of hydrocortisone has been shown to reduce food intake in Sprague-Dawley rats among other adverse effects [24].

Neither compound affected clinical measurements such as body temperature, swelling of the arthritic knee or behaviour of the animals. Many inflammatory cytokines measured from serum, such as TNF-α, IFN-γ, IL-1β, were raised due to arthritis on day 1, but were unaffected by any treatment on day 3. Anti-inflammatory cytokines, like IL-13 and IL-4, that were also induced by Zymosan may contribute to tissue repair and quenching of arthritis within 1–2 weeks. The onset of the anti-inflammatory effect of the study compounds that were administrated on day 1 was likely to be too slow considering the fast progression of arthritis between days 0–1 and rapid spontaneous remission of clinical arthritis within a week. We used a slow-acting calcipotriol formulation to maximise the local treatment effect. In our previous study the same formulation with 0.54–0.22 mg/kg of calcipotriol showed only 8.2–12.6% bioavailability (systemic absorption) in sheep after an intra-articular injection into the knee joint and rapid elimination (with a half-life approximately 1h) [33]. The follow-up period of rats was too short in this study to be able to show possible protective effect of calcipotriol on cartilage.

The ZIA model was chosen for this study since it has been much used previously in studying acute inflammation and pain, sickness behaviour and structural changes of arthritic joints [25–28]. A unilateral model enables the use of the contralateral knee as an internal control. Only male rats were bred for this study to reduce the interference of estrogen, which is known to have a tremendous effect on bone and cartilage health in itself [29–31]. 25-hydroxyvitamin-$D_3$ levels were similar across the study groups and at approximately the same level as described earlier in rats given normal chow [32]; the vitamin D status of the rats was thus not a confounding factor.

There are no previous studies describing ZIA in Sprague-Dawley stock. Based on a pilot study, we reduced the dose of Zymosan from the typical 1–2 mg, causing severe arthritis in Wistar rats [16–18,27–28], to 0.1 mg to get a moderate arthritis in Sprague-Dawley rats. The

A

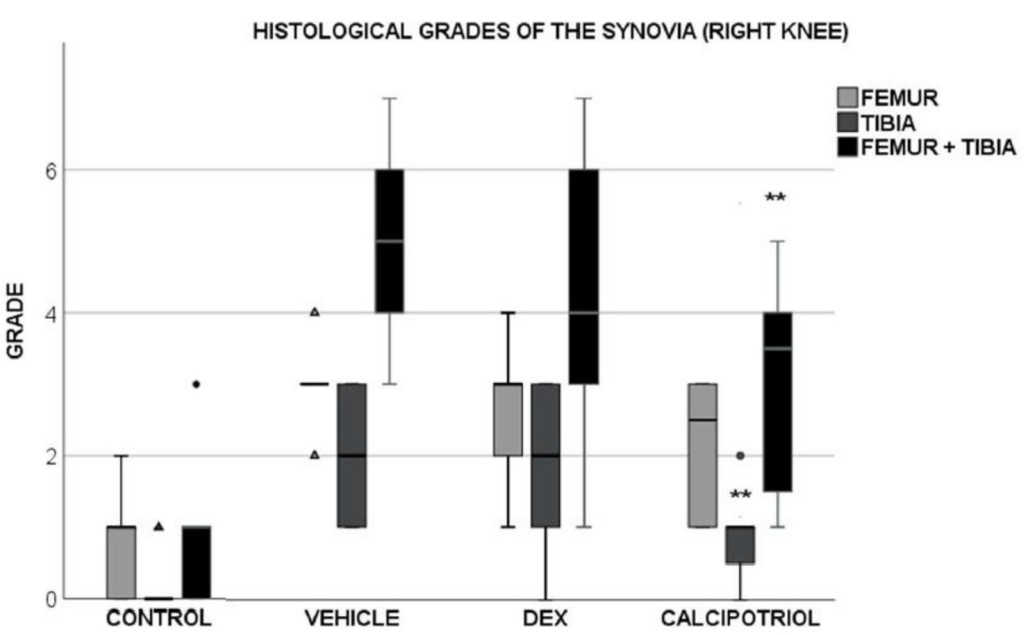

B

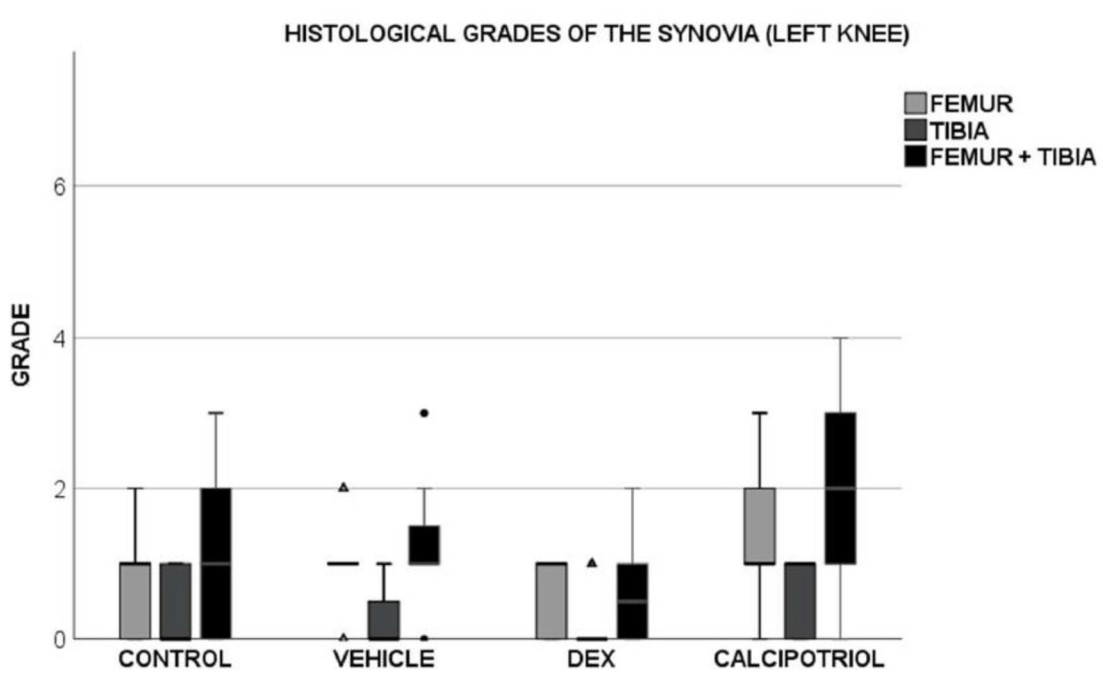

**Fig 4.** Effect of calcipotriol and dexamethasone on the synovitis score on day 8 after arthritis induction in the right, arthritic knee (A) and the left, control knee (B). In the tibial part, the grade was significantly lower in synovia that were exposed to calcipotriol (p = 0.033) (A). The total score (sum of the tibial and femoral part) was also lower in the calcipotriol group compared with the vehicle group (p = 0.041). There were no statistically significant differences between the dexamethasone and vehicle groups. (B) In the left knee, there was no significant difference between the groups (p = ns). Black circles represent outliers that are more than one and a half times the height of the boxes, and triangles represent the extreme outliers that are more than three times the height of the boxes. The number of rats was eight in calcipotriol and vehicle groups, nine in dexamethasone group, and ten in the control group. Statistically significant difference is marked by an asterisk **p<0.05.

age of a rat influenced the severity of arthritis; 3 month-old rats developed fulminant arthritis while 5 month-old rats only moderate arthritis with 0.1 mg of Zymosan. The purpose of the dose reduction of Zymosan was to follow the 3R principles of laboratory animal use and improve the chance to get a treatment response with a single injection of the study compounds.

Considering the fact that we started to treat a fully-developed arthritis on day 1, a relatively big dose of vitamin D analog (0.1 mg/kg of calcipotriol) was used to increase the likelihood of getting a treatment response. We did not find any cytotoxic effects using this and even bigger amount of calcipotriol (0.55 mg/kg) on microscopic evaluation of cartilage in this study or in our previous study with sheep [33]. In an earlier study, a cytotoxic effect on keratinocytes has been detected at ≥10 μM concentration of calcipotriol *in vitro* [34].

Dexamethasone was used as a positive control because of its known anti-inflammatory effects on arthritis [16–18,35]. Side effects of dexamethasone may vary depending on the dose, exposure time and age of rats. Overexposure of dexamethasone in 22-month-old rats for ten days caused long lasting changes in body composition such as reduction of lean body mass and bone mineral content [36]. We speculate that dexamethasone, with water solubility of 0.1 mg/ml compared with 0.6 μg/ml of calcipotriol, might have been absorbed from the synovial space into circulation faster than calcipotriol, which remained in crystals for a longer period of time. In our previous study with sheep, visible amount of calcipotriol crystals were still seen in the synovial fluid sample after two weeks of intra-articular injection of 0.1 mg/kg calcipotriol [33].

The non-fasting glucose levels of the rats in this experiment were higher than fasting plasma reference values of 13-week-old Sprague-Dawley rats (7.2±1.0 mmol/l) [37]. Also, isoflurane anesthesia [38] and moving the cage [39] have been shown to raise plasma glucose.

We noticed a stronger swelling and slightly higher average surface temperatures in the right Zymosan-injected knees compared with the left saline-injected knees, but no differences between the treatment groups. In a previous study with carrageenan-induced arthritis the peak surface temperature (vasodilatation) was detected 4 hours post injection, and the increase in temperature preceded paw swelling [40]. It is possible that the dosing of study compounds was too late as maximal swelling and temperature difference between the arthritic and control knees were already achieved on day 1.

The preliminary work suggests that calcipotriol relieves histological synovitis by inhibiting proliferation of the synovial membrane. Both tibial and total scores were significantly reduced in calcipotriol versus vehicle group, statistical significance could be reached despite the small sample size. The lack of significance on the femoral side is unknown but could potentially be affected by location of calcipotriol crystals lower in the joint space. From a grading perspective, a non-inflamed synovium of controls on the femoral side typically had more than two layers at some spots (gradus 1–2) while the tibial site remained very neat (gradus 0). This might have decreased the variation in grades at the femoral site, leading to slight lack of significance.

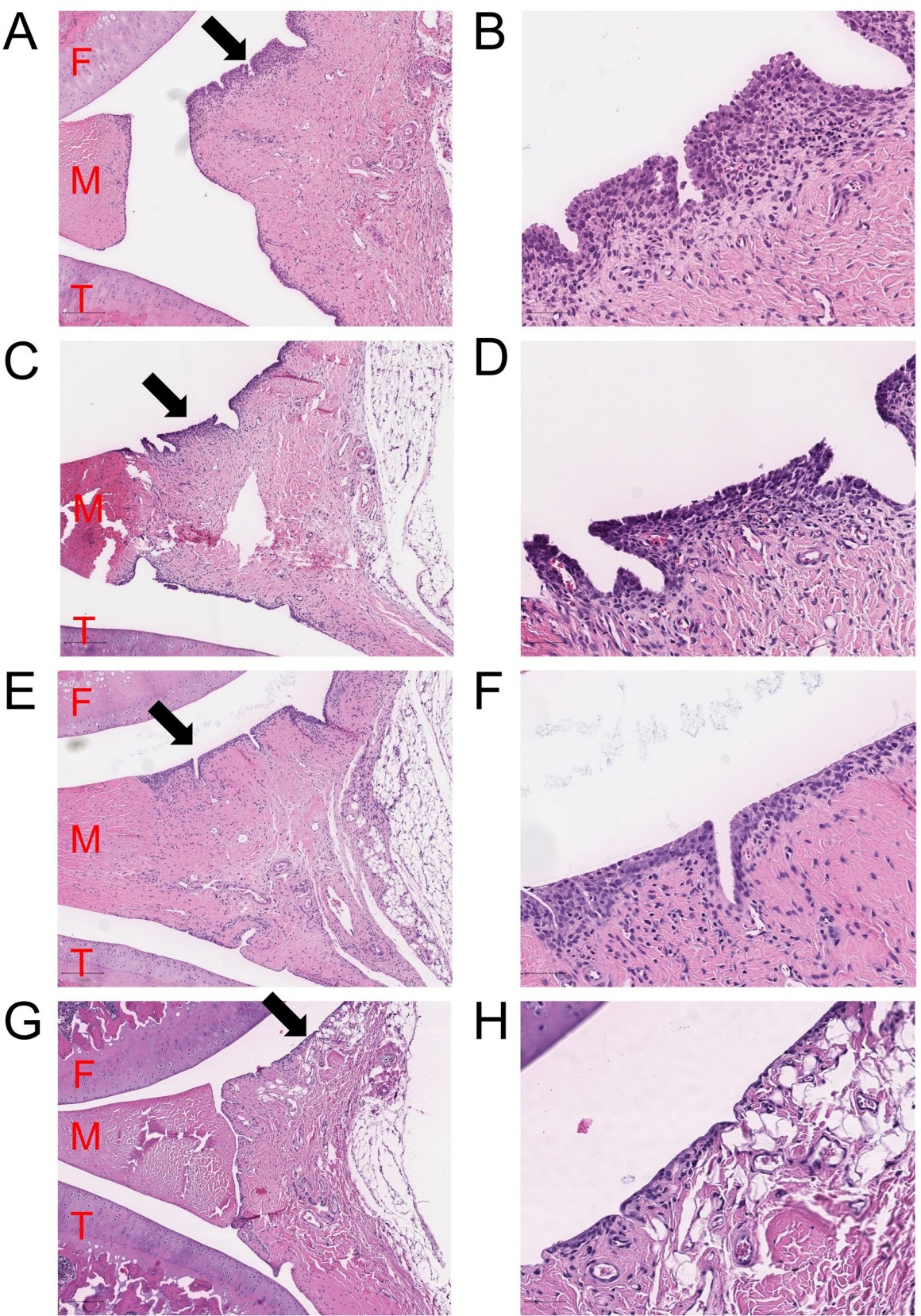

**Fig 5. Histology of the right knees of the arthritic and control rats on day 8.** Synovia as a sagittal view near the midline of the right arthritic knees for vehicle (A-B), dexamethasone (C-D), calcipotriol (E-F) and non-arthritic control (G-H) on day 8. The z-planes were chosen near the midline approaching from the medial side, preferentially patella not visible, and the femur completely visible in the section. The image areas are taken near the posterior part of the joints with 5x magnification on the left and 20x magnification on the right panels. F = femur (cartilage), T = tibia (cartilage), M = medial meniscus. Inflamed and proliferated synovia was observed in all knees injected with Zymosan (A-F). The average score of the synovial grading was smaller in the calcipotriol treated joint compared with vehicle. The dexamethasone and vehicle groups were similar. In non-arthritic control (G-H), the synovial membrane was clearly thinner. Representative images of hematoxylin-eosin staining have been chosen from each group. Extra images presenting the rats with the highest scores of synovial grading in each arthritic group are shown in Supporting Information (S3).

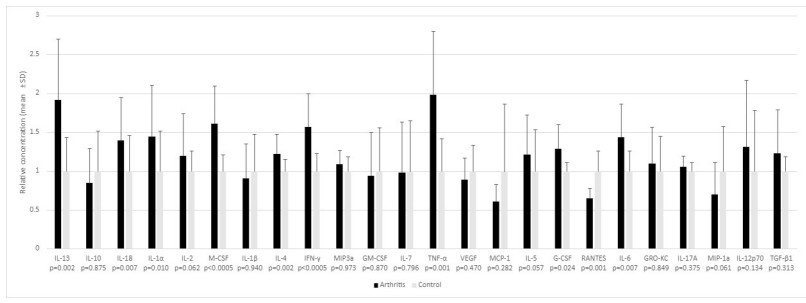

**Fig 6. Serum cytokine levels measured before administration of study compounds on day 1 in arthritic rats (n = 25) and before euthanasia in nonarthritic controls.** (n = 10). A significant increase in many of pro- and anti-inflammatory cytokines caused by acute arthritis was seen on day 1. RANTES was reduced in the arthritis group vs control (p = 0.002). Bars expressed as mean±SD.

## Conclusions

This blinded study in ZIA rats suggests that calcipotriol alleviates synovitis after a single intra-articular dose without any apparent adverse effects. Clinical studies are needed to verify the antiarthritic effect of calcipotriol. In addition, the safety of repeated intra-articular doses of calcipotriol must be tested.

## Supporting information

**S1 Fig. The clinical scores of the rats during the study.** A. The sum of clinical scores on each day during the experiment in different arthritis groups. Comparing the nine-day-sum of all scores, there was no significant difference between the arthritis groups (p = 0.881 in Kruskal-Wallis test). B. The amount of rats according to the nine-day sum score at the end of the experiment. All of the rats had relatively low scores and showed practically no pain behaviour or difficulties in moving. The scores were given daily to each arthritic rat using the scoring sheet in S1 Table. The average scores were minimal, in average <1 every day in every group and thus, the scoring is represented as sums.
(TIF)

**S2 Fig. Examples of strong synovitis findings among the arthritis groups.** From each group, the highest grade was chosen. All images are from posterior recesses with HE staining; 5x magnification on the left, 20x magnification on the right panel. In pictures A and B, a strong synovitis in a vehicle-treated rat is seen. In C-D, a slightly milder synovitis is seen in a calcipotriol-treated rat. Pictures E-F represent a dexamethasone-treated rat with strong synovitis. Also note the thinner synovium in the tibial side of calcipotriol-treated knee (C) compared to other knees (A, E).
(TIF)

**S1 Table. Clinical scoring sheet of the rats, English translation.** The scoring sheet was used for clinical scoring on a daily basis for each arthritic rat. The results are shown in S2 Fig. (DOCX)

## Acknowledgments

We thank the staff of the Oulu Laboratory Animal Center of Oulu University. Minna Savi-lampi is acknowledged for assistance with intra-articular injections and Pia Mäkelä and Erja Tomperi for preparing high-quality histological sections.

## Author Contributions

**Conceptualization:** Johanna A. Huhtakangas, Sakari Laaksonen, Hanna-Marja Voipio, Petri P. Lehenkari.

**Data curation:** Mikko A. J. Finnilä, Jérôme Thevenot.

**Formal analysis:** Jere Huovinen, Olli Vuolteenaho, Jérôme Thevenot.

**Funding acquisition:** Johanna A. Huhtakangas.

**Investigation:** Johanna A. Huhtakangas, Petri P. Lehenkari.

**Methodology:** Jere Huovinen, Sakari Laaksonen, Hanna-Marja Voipio, Mikko A. J. Finnilä, Jérôme Thevenot.

**Project administration:** Johanna A. Huhtakangas, Hanna-Marja Voipio, Petri P. Lehenkari.

**Resources:** Sakari Laaksonen, Hanna-Marja Voipio, Petri P. Lehenkari.

**Software:** Olli Vuolteenaho, Mikko A. J. Finnilä, Jérôme Thevenot.

**Supervision:** Johanna A. Huhtakangas, Hanna-Marja Voipio, Mikko A. J. Finnilä, Jérôme Thevenot, Petri P. Lehenkari.

**Validation:** Jere Huovinen, Hanna-Marja Voipio, Mikko A. J. Finnilä.

**Visualization:** Jere Huovinen, Jérôme Thevenot.

**Writing – original draft:** Johanna A. Huhtakangas, Jere Huovinen, Sakari Laaksonen, Jérôme Thevenot.

**Writing – review & editing:** Johanna A. Huhtakangas, Sakari Laaksonen, Hanna-Marja Voipio, Olli Vuolteenaho, Mikko A. J. Finnilä, Petri P. Lehenkari.

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
