## [Decision Letter · Decision Letter 0]

11 Feb 2021

PONE-D-21-00496

A single intra-articular dose of vitamin D analog calcipotriol alleviates synovitis without adverse effects in rats

PLOS ONE

Dear Dr. Huhtakangas,

Thank you for submitting your manuscript to PLOS ONE. After careful consideration, we feel that it has merit but does not fully meet PLOS ONE’s publication criteria as it currently stands. Therefore, we invite you to submit a revised version of the manuscript that addresses the points raised during the review process.

The manuscript data is interesting, but it needs several careful modifications inline with Reviewers.

We look forward to receiving your revised manuscript.

Kind regards,

Academic Editor

PLOS ONE

Journal Requirements:

Additional Editor Comments:

The manuscript data is interesting, but it needs several modifications inline with Reviewers.

Reviewers' comments:

Reviewer's Responses to Questions

**Comments to the Author**

1. Is the manuscript technically sound, and do the data support the conclusions?

Reviewer #1: Yes

Reviewer #2: Partly

Reviewer #3: Partly

2. Has the statistical analysis been performed appropriately and rigorously? 

Reviewer #1: Yes

Reviewer #2: I Don't Know

Reviewer #3: No

3. Have the authors made all data underlying the findings in their manuscript fully available?

Reviewer #1: Yes

Reviewer #2: Yes

Reviewer #3: No

4. Is the manuscript presented in an intelligible fashion and written in standard English?

Reviewer #1: Yes

Reviewer #2: Yes

Reviewer #3: Yes

5. Review Comments to the Author

Reviewer #1: Dear Editor,

Thank you for inviting me as a reviewer. Following the review of the manuscript, entitled, “A single intra-articular dose of vitamin D analog calcipotriol alleviates synovitis without adverse effects in rats” by Johanna Huhtakangas, Jere Huovinen, Sakari Laaksonen, Hanna-Marja Voipio, Olli Vuolteenaho, Mikko A.J. Finnilä, Jerome Thevenot, Petri P. Lehenkari, I recommend that it should be revised taking into given below suggestions.

1. In the Introduction, line 33-35, “In accordance with this, prophylactic treatment with vitamin D molecules have also shown disease modifying effects in some surgical models of OA in rodents in the onset of OA (10-12) but the beneficial effect was lost in the in the progression phase of OA (10-11)” are little confusing, please rephrase this.

2. Line 35, “beneficial effect was lost in the in the progression phase of OA”, the words “in the” are redundant.

3. In the Introduction, lines 47-48, “The aim of this study was to test hypothesis that calcipotriol that is shown to have anti-inflammatory effects on synoviocytes in vitro (4), would also alleviate arthritis in vivo in rat ZIA model” are little confusing, please rephrase this.

4. Line 110 recheck the spelling of “chemiluminesence” and remove following an extra period.

5. Line 114, the correct word will be “facial expression”.

6. Line 126, “and the rat were always positioned”, the proper verb should be “was” or change the subject word to “rats”.

7. Section “Preparation and analysis of the histological specimen”, line number 154, “the were cut into 5μm sagittal slices with a microtome”, clarify this sentence.

8. Line 175, “175 temperature and knee calibers between days 0 and 1 was studied”, replace the word “was” with “were”.

9. Section “clinical characteristics”, line number 194, “The relative weight loss was greater in the dexamethasone group than in the other arthritis groups on days 2–3 (6% vs. 3-4%, respectively, p<0.05)”, elaborate the cause and consequences of this observation.

10. Line 227,” The data is expressed as mean ± SD.”, replace the “is” with “are”. Amend in the subsequent occurrence too.

11. The whole section “Thermal imaging (starting from line 235)” is quite confusing. Add a few lines to explain the rationale of “The increase in average temperature 240 of the right knee from day 0 to day 1 was 0.74 �C (from 35.02 to 35.76�C, p<0.0005) and the left 241 knee 0.59�C (from 35.04 to 35.63�C, p<0.0005) (right vs. left knee p=0.011), with no significant 242 increase in the surface temperature of the abdomen (from 34.97 to 35.09�C, p=0.356).”

12. Line 286, “Fig 5. Histology the right knees of the arthritic and control rats on day 8”, it should be “Histology of the…….”.

13. Line 298, “on day 1 in arthritic rats versus controls (p=0.002), which may indicate increased consumption”, the period is missing at the end of this sentence.

Thanks,

Muhammad Muaaz Aslam

Reviewer #2: The manuscript provides preliminary work dedicated to study the effect of calcipotriol on synovitis using a Zymosan-induced arthritis (ZIA) rat model. The overall set of data seems well presented and the discussion section in the manuscript is well-written. However, there are a few issues that need to be resolved before the manuscript can be published –

1. The abstract needs to be revised as it neither introduces the field properly, nor it discusses the question which could be answered with the current work. Hence, it fails to provide the importance to the work. Kindly, rewrite the abstract accordingly.

2. Why is the table in Supporting Information S1 empty? Kindly, provide the required data in the table.

3. Please, provide more details on Figure 5. The Z-planes appear to be different in the sections. Kindly, explain how the sectioning was performed in order to reach same planes for analysis. Extra images should be provided as this links to the main conclusion of the study. Quantitation of the thickness of synovial membrane may help.

4. Figure 6 is too large and have excessive number of panels. To simplify, the data can be grouped based on significance levels or type of parameters and the relevant panels can be kept in Figure 6, along with proper description in respective result section. Rest of the data can go to Supporting Information. More elaboration on this huge dataset will be helpful. Most data in the figure 6, appear to be non-significant and the substantially tall error bars make the data less convincing. Kindly, provide clear description on p-values between samples. For instance, the first penal of Figure 6 indicating significant differences among the Day 1 values in IL-13 levels is not convincing. Though, compared to the control it may stand significant. In such case, it should be properly indicated to guide the readers. Also, why the vehicle would induce cytokines? Please, explain.

Overall, the manuscript is interesting and contains ample amount of data. Hence, if authors resolve the above concerns, may turn into a promising publication.

Reviewer #3: Study done by Huhtakangas et. al. entitled “A single intra-articular dose of vitamin D analog calcipotriol alleviates synovitis without adverse effects in rats”, proposed reversal of Zymosan-induced arthritis by single calcipotriol injection without obvious side effects, however, a critical revision is strongly recommended.

There is major confusion with the study time point in this study. Sometimes, author included 0h, 1h, 3h and 8h, whereas in cytokines they presented the data with only 3 time points i.e. 1h, 3h, and 8h. Moreover, Author claimed that they recorded the cytokine measurement prior to compounds treatment and presented data as day 1 data. Also, authors should need to discuss the cellular/molecular processes affected by Zymosan treatment and calcipotriol injection.

Any specific reason to performed under general anesthesia on days 0, 1, 3 and 8 only. It would be great if author explain why they decided to go with this timeline.

Line 34 – Author please elaborate “OA” stands for.

Intro need some editing, especially para 3.

Author should include exact figure number [for example Fig 1A, Fig 1B, … Fig 2A etc].

Although, the authors observed gradual weight loss in all the treated groups, why there is sudden weight loss in dexamethasone group at early stages of time-line. Author should discuss this result.

Table-1 – the presentation of parameter “timing” is bit confusing. Why authors recorded various parameters for the control group only at termination. 25-(OH)2D3 concentration at early phase (e.g. day 0 or day 1) is necessary.

Fig 2, There is mismatch of figure panel sequence and figure legend.

In Fig 2B, Why the control group has wider data distribution along Y-axis as compared to all other groups?

Fig 6 – Authors showed that inflammatory cytokines were increased at day 1 in all Zymosan treated group. As Calcipotriol has antiproliferative and anti-inflammatory effects,

Authors should explain the reason behind the increased expression of both pro- and anti-inflammatory cytokines on day one. Moreover, Authors did not explain on what basis they selected the cytokines to test against the treatment. A table with complete statistical values is required.

Authors reported that VEGF level significantly reduced in calcipotriol group, however, error bars in the graph (Fig 6) is not going along with the statement.

Authors should present a dose dependent result that supports their selection of Zymosan dosing of 0.1 mg.

Authors has presented format table of clinical scoring but it is not clear that how the scoring distributed. Marely presenting the format sheet does not make any sense. Additionally, Fig S2 need better presentation. X- and Y-axis are missing and strongly needed to make figure understandable.

6. PLOS authors have the option to publish the peer review history of their article (what does this mean?). If published, this will include your full peer review and any attached files.

Reviewer #1: No

Reviewer #2: **Yes: **Deepak Chhangani

Reviewer #3: No

---

## [Author Response · Author response to Decision Letter 0]

10 Mar 2021

Please find the answers attached as a separate file.

---

## [Decision Letter · Decision Letter 1]

6 Apr 2021

A single intra-articular dose of vitamin D analog calcipotriol alleviates synovitis without adverse effects in rats

PONE-D-21-00496R1

Dear Dr. Huhtakangas,

We’re pleased to inform you that your manuscript has been judged scientifically suitable for publication and will be formally accepted for publication once it meets all outstanding technical requirements.

Kind regards,

Rajakumar Anbazhagan, Ph. D.

Academic Editor

PLOS ONE

Additional Editor Comments (optional):

Reviewers' comments:

Reviewer's Responses to Questions

**Comments to the Author**

1. If the authors have adequately addressed your comments raised in a previous round of review and you feel that this manuscript is now acceptable for publication, you may indicate that here to bypass the “Comments to the Author” section, enter your conflict of interest statement in the “Confidential to Editor” section, and submit your "Accept" recommendation.

Reviewer #3: All comments have been addressed

Reviewer #4: All comments have been addressed

2. Is the manuscript technically sound, and do the data support the conclusions?

Reviewer #3: Yes

Reviewer #4: Yes

3. Has the statistical analysis been performed appropriately and rigorously? 

Reviewer #3: Yes

Reviewer #4: Yes

4. Have the authors made all data underlying the findings in their manuscript fully available?

Reviewer #3: Yes

Reviewer #4: Yes

5. Is the manuscript presented in an intelligible fashion and written in standard English?

Reviewer #3: Yes

Reviewer #4: Yes

6. Review Comments to the Author

Reviewer #3: (No Response)

Reviewer #4: The authors have addressed the reviewers comments and the revised version is technically sound and written in intelligible fashion.

7. PLOS authors have the option to publish the peer review history of their article (what does this mean?). If published, this will include your full peer review and any attached files.

Reviewer #3: No

Reviewer #4: No

---

## [Editor Report · Acceptance letter]

8 Apr 2021

PONE-D-21-00496R1 

A single intra-articular dose of vitamin D analog calcipotriol alleviates synovitis without adverse effects in rats 

Dear Dr. Huhtakangas:

I'm pleased to inform you that your manuscript has been deemed suitable for publication in PLOS ONE. Congratulations! Your manuscript is now with our production department. 

Kind regards, 

on behalf of

Dr. Rajakumar Anbazhagan 

Academic Editor

PLOS ONE